# Demographic responses of a threatened, low-density ungulate to annual variation in meteorological and phenological conditions

**Craig A. DeMars**[1]*, **Sophie Gilbert**[2], **Robert Serrouya**[1], **Allicia P. Kelly**[3], **Nicholas C. Larter**[4], **Dave Hervieux**[5], **Stan Boutin**[6]

**1** Caribou Monitoring Unit, Alberta Biodiversity Monitoring Institute, Edmonton, AB, Canada, **2** Department of Fish & Wildlife Sciences, University of Idaho, Moscow, ID, United States of America, **3** Department of Environment and Natural Resources, Government of Northwest Territories, Fort Smith, NT, Canada, **4** Department of Environment and Natural Resources (retired), Government of Northwest Territories, Fort Simpson, NT, Canada, **5** Alberta Environment and Parks, Grande Prairie, AB, Canada, **6** Department of Biological Sciences, University of Alberta, Edmonton, AB, Canada

* cdemars@ualberta.ca

**Data Availability Statement:** Data are available in Dryad: datadryad.org; data doi: 10.5061/dryad. stqjq2c3g.

## Abstract

As global climate change progresses, wildlife management will benefit from knowledge of demographic responses to climatic variation, particularly for species already endangered by other stressors. In Canada, climate change is expected to increasingly impact populations of threatened woodland caribou (*Rangifer tarandus caribou*) and much focus has been placed on how a warming climate has potentially facilitated the northward expansion of apparent competitors and novel predators. Climate change, however, may also exert more direct effects on caribou populations that are not mediated by predation. These effects include meteorological changes that influence resource availability and energy expenditure. Research on other ungulates suggests that climatic variation may have minimal impact on low-density populations such as woodland caribou because per-capita resources may remain sufficient even in "bad" years. We evaluated this prediction using demographic data from 21 populations in western Canada that were monitored for various intervals between 1994 and 2015. We specifically assessed whether juvenile recruitment and adult female survival were correlated with annual variation in meteorological metrics and plant phenology. Against expectations, we found that both vital rates appeared to be influenced by annual climatic variation. Juvenile recruitment was primarily correlated with variation in phenological conditions in the year prior to birth. Adult female survival was more strongly correlated with meteorological conditions and declined during colder, more variable winters. These responses may be influenced by the life history of woodland caribou, which reside in low-productivity refugia where small climatic changes may result in changes to resources that are sufficient to elicit strong demographic effects. Across all models, explained variation in vital rates was low, suggesting that other factors had greater influence on caribou demography. Nonetheless, given the declining trajectories of many woodland caribou populations, our results highlight the increased relevance of recovery actions when adverse climatic conditions are likely to negatively affect caribou demography.

**Funding:** Partial funding for this work was provided to CAD, SG and RS by the British Columbia Oil and Gas Research and Innovation Society (www.bcogris.ca). The funders did not play any role in the study design, data collection and analysis, decision to publish or preparation of the manuscript. In-kind contributions of caribou demographic data were provided by the governments of British Columbia, Alberta, and the Northwest Territories. None of these governments influenced the design, data analysis, manuscript preparation or the decision to publish.

**Competing interests:** The authors have declared that no competing interests exist.

## 1. Introduction

Climate exerts direct and indirect effects on a species' population dynamics [1–3] and is a key determinant of a species' distribution [4]. Direct climate effects include those related to temperature and precipitation, which dictate seasonal and annual resource availability and energy expenditure, thereby impacting survival and reproductive rates [2, 5]. Climate can also impact a species' demography by altering interactions within and among trophic levels, affecting key demographic processes such as predation [6, 7] and competition [8]. Despite the variable nature of climate effects, species are expected to be adapted to the normal climatic variation of local environments [9]. Current and predicted rates of global climate change, however, will likely challenge the adaptive potential of many species [10] and, consequently, an increased emphasis has been placed on understanding how species respond to varying climatic conditions [11, 12].

Among terrestrial vertebrates, climate effects on ungulate population dynamics have been relatively well-studied, particularly in temperate regions (for reviews see [1, 13, 14]). For the most part, studies have considered climate effects as those related to the variation and/or directional trend in annual meteorological or phenological conditions [7, 15, 16] and we adopt that convention here. Assessments of demographic impacts from these changing conditions have usually focused on juvenile recruitment and/or adult female survival because of their high influence on ungulate demography [17, 18]. Juvenile recruitment represents a combination of rates—pregnancy, fetal survival to parturition and juvenile survival—and each rate can be influenced by climate effects. For example, climate effects can influence late summer and early autumn range conditions, which can impact pregnancy rates and resource accumulation necessary for over-winter fetal survival [19, 20]. Climate effects during winter and early spring can impact fetal survival by affecting maternal energy expenditures and consequently allocation of resources to the fetus [21]. These effects on maternal expenditures and resource allocation can also affect the birth weight of offspring [20, 22], which influences juvenile survival as smaller offspring generally have lower survival probabilities [16, 23]. Juvenile survival is further influenced by summer/autumn range conditions and winter severity as these time periods influence the accumulation and depletion of body reserves critical to over-winter survival [20, 24]. Over-winter survival of adult females is also affected by similar mechanisms (i.e., resource accumulation and reserve depletion; [20]). In addition to these physiological pathways, climate can impact ungulate demography by interacting with ecological processes such as predation, particularly during winter where variation in winter severity has been shown to alter predation risk for both juveniles and adult females [7, 25]. The above examples illustrate relatively direct climate relationships, yet juvenile recruitment and adult female survival can also be influenced by lagged climate effects; for instance, severe winters may limit pregnancy rates and influence juvenile recruitment in the following year [20, 26, 27].

Although it is well-established that climatic variation impacts ungulate demography, there are uncertainties as to the exact nature of many climate-demography relationships, making generalizations difficult [28]. Generalizing across species is particularly problematic as differences in life-history traits can cause species to vary in their response to similar climatic effects. For example, Wilmers et al. [7] reported that increasing snow depth interacted with predation to lower survival in elk (*Cervus elaphus*) whereas increasing snow depth had a weak positive effect on adult female survival in the boreal ecotype of woodland caribou (*Rangifer tarandus caribou*; [29]). Responses to climatic variation may also be variable within species as typified by contrasting climate-demographic relationships reported for caribou and reindeer. Examples include Hegel et al. [30] finding that climate effects (temperature and precipitation) in winter best explained juvenile recruitment in the northern ecotype of woodland caribou

whereas Chen et al. [31] reported that recruitment variability in barren-ground caribou (*Rangifer tarandus groenlandicus*) was best explained by summer range conditions. Joly et al. [32] also showed that climatic conditions indexed by the Pacific Decadal Oscillation had contrasting effects on the growth rates of two caribou populations in Alaska. Similarly, Tyler et al. [33] and Hansen et al. [34] reported contrasting effects of warmer winters on growth rates of spatially separated populations of Svalbard reindeer (*Rangifer tarandus platyrhynchus*). Together, these examples suggest that although general trends may be apparent (e.g., recruitment is more sensitive than adult female survival; [1, 17]), ungulate responses to changing climatic conditions will likely be context- and species-specific.

Understanding climate-demographic relationships is particularly important for ungulate species that are rare and/or endangered because climate effects may be interactive with, or additive to, other population stressors, potentially influencing the selection and/or efficacy of conservation actions [12]. Here, we evaluated the potential influence of climate effects on the demography of woodland caribou, which are comprised of various ecotypes that are currently listed as either endangered, threatened or of special concern in Canada. Population declines of woodland caribou have been primarily attributed to increasing predation ultimately facilitated by human-mediated landscape alteration and, potentially, climate change [35–37]. Increasing predation from climate change has been predominantly linked to the expansion of other ungulate species (e.g. white-tailed deer [*Odocoileus virginianus*]) into caribou range, which increases the abundance of predators that incidentally prey on caribou [36, 38]. Climatic conditions, however, may affect woodland caribou populations via some of the more direct mechanisms described previously. Understanding and quantifying the potential effects of these direct demographic linkages is needed to more fully inform conservation strategies for woodland caribou, particularly given that these caribou are currently the focus of concerted—and ultimately expensive—conservation efforts [37, 39].

To date, few studies have investigated direct climate-related effects on the demography of woodland caribou and all have been restricted to a few discrete areas within the broad and varied geographic distribution of these caribou (e.g., [6, 29, 30, 40]); consequently, the demographic responses of most woodland populations to climatic variation remains largely unknown. This knowledge gap is perhaps surprising for an iconic, threatened animal, but one potential reason for the lack of research in this area is that prevailing literature predicts weak or muted demographic responses of woodland caribou to climatic variation. Within most of their distribution, woodland caribou occur at low densities (e.g. <0.05 caribou/km$^2$; [41, 42]) and their populations are thought to be regulated by predation [41, 43, 44] as densities can exceed 1/km$^2$ in predator-free environments [45]. Populations with these characteristics are likely to have weak responses to climatic variation—because the strength of density dependence is low [46]—and may be resilient even to extreme stochastic weather events as the per-capita resource availability after such events may still be sufficient to sustain low-density populations [5]. Much of the research supporting this low-density/low-impact prediction, however, has been conducted on populations that have relatively wide fluctuations in their densities and/or are not regulated by predation {e.g., [5, 47]).

Within this context, we assessed for direct climate effects on the demography of 21 populations of threatened woodland caribou situated in western Canada. We characterized climate effects using meteorological variables (e.g., temperature and precipitation) and variables representing annual variation in plant phenology, which can index changes in forage quality and quantity for ungulates [48]. We related these climate variables to annual estimates of adult female survival and juvenile recruitment in each population. We specifically evaluated the low-density/low-impact prediction, expecting minimal to no effects on caribou demography given that many of these populations are small and rapidly declining [37, 49] and their low densities

would therefore result in only weak competition for resources even in 'bad' climatic years. If climatic variation did have an observable influence on caribou demography, we expected such effects would be most evident in juvenile recruitment as it is the more variable rate [17, 18]. Across all caribou ranges, we assessed for temporal trends in each climatic variable and discuss our results in the context of global climate change and the potential implications for current and future conservation strategies for woodland caribou.

## 2. Methods

### 2.1. Caribou demographic data

We used demographic data from 21 local populations of woodland caribou (hereafter, populations), which are defined as discrete groups of interacting caribou subject to similar biotic and abiotic processes because group members occupy a shared geographic area (i.e., a range; Fig 1; [37, 50]). Of the 21 populations, eighteen were from the boreal ecotype (Designatable Unit 6) and three were from the central mountain ecotype (Designatable Unit 8; [51]). Data delineating ranges were provided by provincial and territorial governments; consequently, range boundaries partially reflect jurisdictional borders. Because woodland caribou are difficult to count, precise estimates of population size and/or density are lacking for most populations, but all are considered low-density and not expected to exceed approximately 0.05 caribou/km$^2$ [37, 41, 42, 49].

We assessed caribou response to varying meteorological and phenological conditions using two demographic rates: calf-adult female (CAF) ratios, which index juvenile recruitment [18], and adult female survival (AFS). These population-specific demographic data were provided by provincial and territorial governments and were collected between 1994 and 2015, although the monitoring interval varied among populations ($\bar{x}$ = 13.4 years, range: 4–22). CAF ratios were estimated from aerial surveys conducted in March (see S1 Appendix for yearly estimates). These surveys recorded the total number of calves and adult females observed ($\bar{x}$ = 147.3 adult females observed/caribou range/year [range: 11–1288]). Because our focus was not on estimating population growth *per se*, CAF ratios were not adjusted to reflect the number of female calves to the total number of females across all age classes [18]. For AFS, monitoring data from VHF- or GPS-collared adult females ($\geq$2 years old; exact ages on capture are unknown) in each population were used to derive estimates of annual survival rates ($\bar{x}$ = 33.7 adult females monitored/caribou range/year [range: 8–115]; see S1 Appendix for yearly estimates). For VHF-collared females, survival status was determined by aerial telemetry flights conducted 4–12 times per year [52, 53], a monitoring frequency found to produce unbiased survival estimates [54]. Annual rates of AFS for each population were estimated using the Kaplan-Meier method in a staggered entry design [52, 53, 55]. The monitoring interval for estimating survival differed slightly among jurisdictions, with Alberta using a monitoring year of May 1–April 30 while British Columbia and the Northwest Territories used April 1–March 31. For the Chinchaga population where Alberta and British Columbia were independently monitoring demographic rates, we combined data for CAF ratios (i.e., summed the total number of calves and adult females observed across jurisdictions) and derived a combined estimate of AFS using means weighted by sample sizes from each province and year. Across populations, CAF and AFS were generally uncorrelated (Spearman's correlation coefficient [$r$] = -0.01) and AFS was more highly correlated ($r$ = 0.92) with a population's growth rate than CAF ($r$ = 0.38; S1 Appendix).

The total number of monitoring years differed among populations and by demographic rate. In general, CAF ratios were monitored for a longer period (277 population-years) than AFS (268 population-years). As a result, the mean per-population number of years monitored

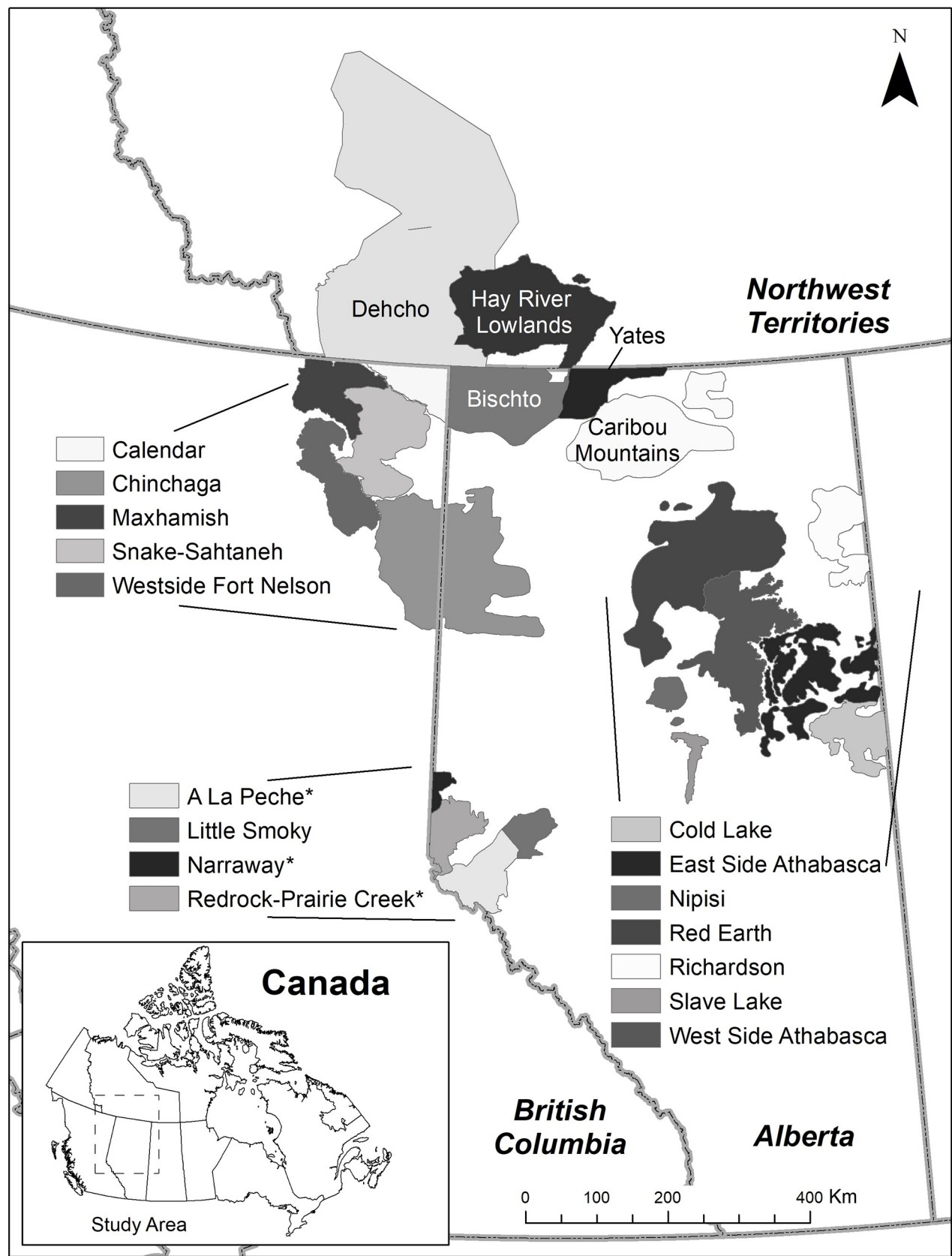

**Fig 1. Range boundaries for 21 populations of woodland caribou in western Canada.** Calf-adult female ratios and adult female survival were monitored annually for various periods between 1994 and 2015 within each range. Asterisk (*) indicates populations of central mountain caribou; all other populations are the boreal ecotype.

was higher for CAF ratios ($\bar{x}$ = 13.2; range: 4–22) than for AFS ($\bar{x}$ = 12.8; range: 3–22). Jurisdiction also influenced monitoring timespans with all BC populations having four years of CAF ratios and three years of AFS data while all Alberta and NWT populations had at least eight years of both demographic rates.

## 2.2. Meteorological and phenological data

We evaluated a suite of meteorological and phenological variables to potentially explain variation in caribou demographic rates. These variables included seasonal means of minimum and maximum temperatures, seasonal accumulations of snowfall or rainfall, the number of freeze-thaw events in a given winter and metrics representing temporal changes in forage quality and availability, which were derived from normalized difference vegetation index (NDVI) data [48]. Because the potential effects of these variables on caribou demography may be lagged or cumulative over years [20, 26], we considered lagged effects up to three years for all explanatory variables. We also assessed for temporal trends in each climate-related variable from 1990–2015 within each caribou range (see *Data Analysis*).

We developed seasonal meteorological variables using Daymet daily surface weather data (1-km resolution; data range: 1980–most current full calendar year) available from the National Aeronautics and Space Administration's (NASA) Oak Ridge National Laboratory (https://doi.org/10.3334/ORNLDAAC/1328; S2 Appendix). For our analyses, we accessed daily estimates of minimum temperature, maximum temperature, precipitation and snow water equivalent (SWE). We used these variables to first delineate two seasonal periods: a thermologically-based growing season and a snow season. We defined the growing season as the annual period when the daily minimum temperature—averaged across a population's range—consistently exceeded 0˚C (i.e., the period between the last spring freeze and the first autumn freeze; S2 Appendix). For the three populations situated in montane regions (Narraway, A La Peche, and Redrock-Prairie Creek), we relaxed this threshold to -3˚C as montane vegetation are adapted to growing in the generally colder conditions found at higher elevations. For each population-year, we developed the following growing season variables: growing season length (in days), growing season start and end dates (Julian day), mean and standard deviation (SD) of the daily maximum temperature, mean and SD of the daily minimum temperature, and cumulative precipitation.

We used a similar approach to develop variables for the snow season. Within each population's range, we used SWE data to calculate the proportion of raster cells that were snow-free each day of the year (S2 Appendix). We identified the start of the snow season by searching these data for an inflection point occurring between 1 October and 31 December, an interval that excluded occasional snow storms in September that covered the landscape then subsequently melted (i.e., we wanted to identify the start of continuous snow cover). We identified the end of the snow season by searching for an inflection point in the time series occurring between 1 January and 30 June of the following year. For each population-year, we developed the following variables: snow season length (in days), snow season start and end dates (Julian day), mean and standard deviation (SD) of the maximum daily temperature, mean and SD of the daily minimum temperature, mean and SD of the daily SWE (in mm; an index of snow depth on the ground), cumulative snow in March (an index of late winter snow depth) and cumulative SWE across the snow season.

We also included the number of freeze-thaw events in snow season analyses. Such events have been shown to negatively affect ungulate demography, primarily because icing can restrict access to forage [34]. We estimated the number of freeze-thaw events in a given winter using data from the Freeze/Thaw Earth System Data Record (25-km resolution; data span 1979–2015; http://freezethaw.ntsg.umt.edu/). These data measure the number of thaw events (units = weeks/year) during a winter (November–April), excluding "transition" events (i.e., autumn freeze-up and spring thaw periods), and only in areas with snow cover. Thaw events were determined by classifying temporal changes in the time-series of microwave brightness temperature observations remotely collected by global satellites.

Annual variation in weather can create annual variation in plant phenology, which can index temporal changes in forage quantity and quality for ungulates [48]. To evaluate the effect of changing plant phenology on caribou demography, we used NDVI data to develop a suite of phenologically-based variables. For pre-2000, we used NDVI data derived from Advanced Very High Resolution Radiometer (AVHRR) imagery available from NASA (8-km resolution; produced bi-monthly; https://ecocast.arc.nasa.gov/data/pub/gimms/3g.v0/). For 2000–2015, we used NDVI data developed from the Terra Moderate Resolution Imaging Spectroradiometer (MODIS), which are also available from NASA (250-m resolution; produced from 16-day composite imagery; Vegetation Indices [MOD13Q1] Version 6; https://lpdaac.usgs.gov/products/mod13q1v006/). Although the two data sets have differing resolutions, this difference is likely not problematic because we calculated mean NDVI values over a large spatial extent (i.e., a population's range) and comparisons of AVHRR and MODIS data have found high levels of agreement [56].

We developed phenological variables following an approach similar to that used for the meteorological variables. For each time-step, we first calculated a mean NDVI value for each caribou range. We then delineated a phenologically-defined growing season by fitting a smoothed curve to these NDVI values for each population-year (S2 Appendix). We identified greenup (the start of the growing season) and senescence (the end of the growing season) using the 55th difference percentile between the maximum and minimum modelled NDVI values on each side of the curve. Within the growing season, we also recorded the following phenological variables: growing season length, the highest modelled NDVI value, the date when this value was reached, and the integrated NDVI (iNDVI), which is the sum of all NDVI values across the growing season [48].

## 2.3. Data analysis

We assessed for temporal trends in each meteorological and phenological variable within each caribou range using linear mixed-effects models, which account for the hierarchical structure of the data (i.e., repeated annual measures for each population). To facilitate trend comparisons, we scaled each climate-related variable prior to model estimation. For each model, we specified caribou range as a random grouping factor (i.e. a random intercept) and 'Year' as a random slope, which allows for potential trends to vary by range. We also specified a continuous autoregressive process (AR(1)) as the correlation structure.

We used a two-stage approach to build regression models for evaluating meteorological and phenological effects on caribou demography. In the first stage, we used principal component analysis (PCA) to summarize the variables within each season (i.e., thermological growing season, phenological growing season, and snow season) and these summaries were developed for the monitoring year and for the three years prior (lagged effects). This dimensionality reduction was necessary because the small sample size of caribou populations (*n* = 21) limited the fitting of complex multi-variate regression models. All climate-related

variables were scaled prior to their input into each PCA to facilitate comparisons of each variable's relative contribution. PCA summaries were easily interpretable as the input variables described temporal characteristics of the season (i.e., start, end and length) and within-season meteorological or phenological characteristics, yielding, for example, contrasts such as long, hot growing seasons versus short, cool ones. In general, the first two principal components (PCs) explained most of the variance (>65%) within each suite of variables and we used these PCs as the primary explanatory variables in subsequent regression models.

In the second stage of analysis, we related caribou demographic rates to the seasonal PCs by fitting generalized linear mixed-effects models (GLMMs). Following Hegel et al. [30], we specified year and population as crossed random grouping factors for all models to capture baseline differences among populations and years. We also created a "trend" variable to account for possible long-term trends in both demographic rates and thus potentially increase discrimination of true meteorological or phenological effects. To maintain population as the primary sampling unit and generate appropriate standard errors [57], we specified all PC variables and, where possible (see further below), trend variables as random slopes within GLMMs, thereby generating population-specific coefficients. These models took the form

$$CAF \text{ or } AFS = \beta_0 + \beta_1 x_{1ijk} + \beta_2 Trend_{ijk} + \gamma_{0j} + \gamma_{0k} + \gamma_{nij} x_{nij}$$

where $\beta_0$ is the fixed-effect intercept, $\beta_n$ is the fixed-effect—or population mean—coefficient for covariate $x_n$, $\gamma_{0j}$ is the random intercept for population $j$, $\gamma_{0k}$ is the random intercept for year $k$, and $\gamma_{nij}$ is the random slope (or coefficient) of covariate $x_n$ for population $j$. Given this formulation and our sample size ($n = 21$ populations), we restricted models to two fixed-effect variables (one PC variable and the trend variable).

Modelled distributions for GLMMs depended on the demographic rate. For AFS models, we specified a beta distribution with a logit link because survival rates could potentially range from 0–1.0. Prior to model fitting, we transformed the survival data using the following formula [58]

$$(y * (n - 1) + 0.5)/n$$

where $n$ is the sample size. This transformation was necessary because beta regression models require that response values be >0 and <1 and AFS in some population-years was 1.0 [59]. In all AFS models, we specified the maximum number of radio-collared females in a given year (e.g. 1 April–31 March) as a sample weight. For CAF models, we used a binomial distribution with a logit link and the number of adult females observed on each survey was specified as a sample weight. We used a binomial distribution because it is extremely rare for female woodland caribou to have more than one calf per season and thus CAF ratios always fall within the 0–1 interval. Subsequent diagnostic testing of CAF models suggested a high level of overdispersion (S3 Appendix), which can potentially bias model coefficients and their variance estimates. To account for overdispersion, we added an observation-level random effect to all CAF models. The inclusion of this additional parameter, however, prevented the inclusion of 'trend' as a random slope as models with this higher complexity generally failed to converge. 'Trend' was therefore specified as a fixed-effect only in CAF models. We further tested demographic models for potential impacts from spatial autocorrelation, given the apparent spatial clustering of caribou ranges (Fig 1). Inspection of variograms using model residuals suggested low correlation among adjacent ranges and the fitting of models with spatial covariance structures did not improve model performance (S3 Appendix).

Across all demographic models, we assessed goodness-of-fit (S3 Appendix) and evaluated effect sizes and their 95% CIs. We also calculated two $R^2$ statistics to summarize model fit: the

marginal $R^2_{GLMM(m)}$, which is the explained variation of the fixed effects, and the conditional $R^2_{GLMM(c)}$, which is the explained variation provided by both the fixed and random effects [60]. For climate-related variables showing evidence of a strong influence on caribou demography (i.e., a fixed-effect with a 95% CI not overlapping zero), we evaluated for potential latitudinal trends by regressing the population-specific coefficients (the random slopes) against the latitude of centroids estimated for each population's range. Latitude variables were scaled prior to model estimation.

All analyses were performed in R, version 3.6.3 (see S4 Appendix for the specific packages used and associated references).

## 3. Results

### 3.1. Temporal trends in meteorological and phenological variables

Temporal trends were apparent in a multitude of the meteorological and phenological variables (Table 1). For the meteorological growing season, trends were evident for later start and end dates to the growing season, higher mean maximum daily temperatures and lower accumulations of precipitation (all $p \leq 0.01$). For the phenological growing season, trends were evident for a longer growing season, primarily driven by later senescence (both $p \leq 0.01$). Productivity also appeared to slightly decrease and maximum productivity generally moved toward a later date (both $p \leq 0.01$). Trends for a later ending to the growing season translated to a trend for a later start to the snow season ($p < 0.01$). Other trends during the snow season included higher minimum temperatures, less variability in daily maximum and minimum temperatures and lower snow snowfall accumulations across the season (all $p \leq 0.02$).

### 3.2. Demographic responses of caribou to meteorological and phenological variation

**3.2.1. Meteorological growing season.**   The first two PCs of the meteorological growing season were primarily influenced by season length, the end date of the season, and the mean maximum and minimum temperatures (Fig 2; see S5 Appendix for further details on PCA results). The first PCs generally described longer growing seasons with warmer but more variable minimum temperatures. The growing seasons characterized by the second PCs varied slightly depending on the time period. For the monitoring year and two- and three-year lags, the second PCs described cooler and wetter growing seasons whereas the second PC for one-year lags had different directional loadings and consequently described growing seasons that were warmer and drier.

The longer and warmer growing seasons described by the first PCs influenced the two demographic rates differently. CAF trended toward a negative relationship with this type of season in each of the four time periods considered (birth year and one, two, or three years prior; Fig 2). Among these periods, CAF was most influenced by longer, warmer growing seasons when they occurred in the year prior to birth (β = -0.17, 95% CI: -0.32, -0.02, Fig 2). This relationship yielded predicted values of CAF that decreased from 0.36 to 0.05 as the growing season lengthened and warmed within the range of modelled values (Fig 5). Population-specific coefficients for this relationship did not show a latitudinal trend ($p = 0.19$). In contrast to CAF, AFS had a positive correlation when longer, warmer growing seasons occurred one year prior to the monitoring year (β = 0.16, 95% CI: 0.01, 0.31) and was less influenced when these growing seasons occurred in other time periods. For the one-year lagged relationship, predicted values for AFS increased from 0.66 to 0.95 across the range of modelled values (Fig 5) and population-specific coefficients did not show a latitudinal trend ($p = 0.47$).

**Table 1. Estimated temporal trends in variables used to assess the demographic response of woodland caribou to annual meteorological and phenological variation.**

| Season | Response Variable | β (Year) | SE | p |
|---|---|---|---|---|
| Growing (Meteorological) | | | | |
| | Growing Season Length | -0.003 | 0.005 | 0.51 |
| | Growing Season Start | 0.018 | 0.005 | <0.01 |
| | Growing Season End | 0.016 | 0.006 | 0.01 |
| | Mean Maximum Temperature | 0.031 | 0.005 | <0.01 |
| | Mean Minimum Temperature | 0.005 | 0.004 | 0.30 |
| | SD[1] Maximum Temperature | -0.004 | 0.006 | 0.49 |
| | SD Maximum Temperature | 0.003 | 0.004 | 0.41 |
| | Cumulative Precipitation | -0.027 | 0.006 | <0.01 |
| Growing (Phenological) | | | | |
| | Growing Season Length | 0.053 | 0.005 | <0.01 |
| | Greenup | -0.010 | 0.005 | 0.06 |
| | Senescence | 0.058 | 0.005 | <0.01 |
| | iNDVI[2] | -0.027 | 0.005 | <0.01 |
| | Maximum NDVI Value | -0.027 | 0.004 | <0.01 |
| | Date of Maximum NDVI Value | 0.024 | 0.005 | <0.01 |
| Snow | | | | |
| | Snow Season Length | -0.009 | 0.005 | 0.09 |
| | Snow Season Start | 0.027 | 0.005 | <0.01 |
| | Snow Season End | 0.007 | 0.006 | 0.22 |
| | Mean Maximum Temperature | 0.002 | 0.003 | 0.50 |
| | Mean Minimum Temperature | 0.011 | 0.004 | <0.01 |
| | SD Maximum Temperature | -0.018 | 0.004 | <0.01 |
| | SD Minimum Temperature | -0.035 | 0.005 | <0.01 |
| | Mean Daily SWE[3] | -0.014 | 0.006 | 0.02 |
| | SD Daily SWE | -0.010 | 0.006 | 0.07 |
| | Cumulative SWE | -0.015 | 0.006 | 0.01 |
| | March SWE | -0.008 | 0.006 | 0.15 |
| | Freeze-Thaw Events | -0.017 | 0.005 | <0.01 |

Shown are model coefficients (β) and their standard errors (SE) for 'Year' from linear mixed-effects models. All response variables were scaled prior to model estimation.

[1] SD = standard deviation.

[2] integrated normalized difference vegetation index.

[3] SWE = snow water equivalents.

Growing seasons characterized by the second PCs also had varying influences on CAF and AFS (Fig 2). Cooler, wetter growing seasons generally had minimal influence on CAF except for a weak negative correlation when these seasons occurred on a two-year lag (β = -0.09, 95% CI: -0.20, 0.01). These type of seasons had a stronger, negative correlation with AFS but only three years after they occurred (β = -0.14, 95% CI: -0.24, -0.05). Predicted values from three-year lagged relationship suggested a decline in AFS from 0.91 to 0.76 as growing seasons became increasingly cooler and wetter within the range of modelled values (Fig 5). This relationship also skewed toward a weak negative trend with latitude (scaled β = -0.09, 95% CI: -0.19, 0.01). Neither CAF nor AFS had any strong correlation with the warmer, drier growing seasons characterized by the second PCs for one-year lags.

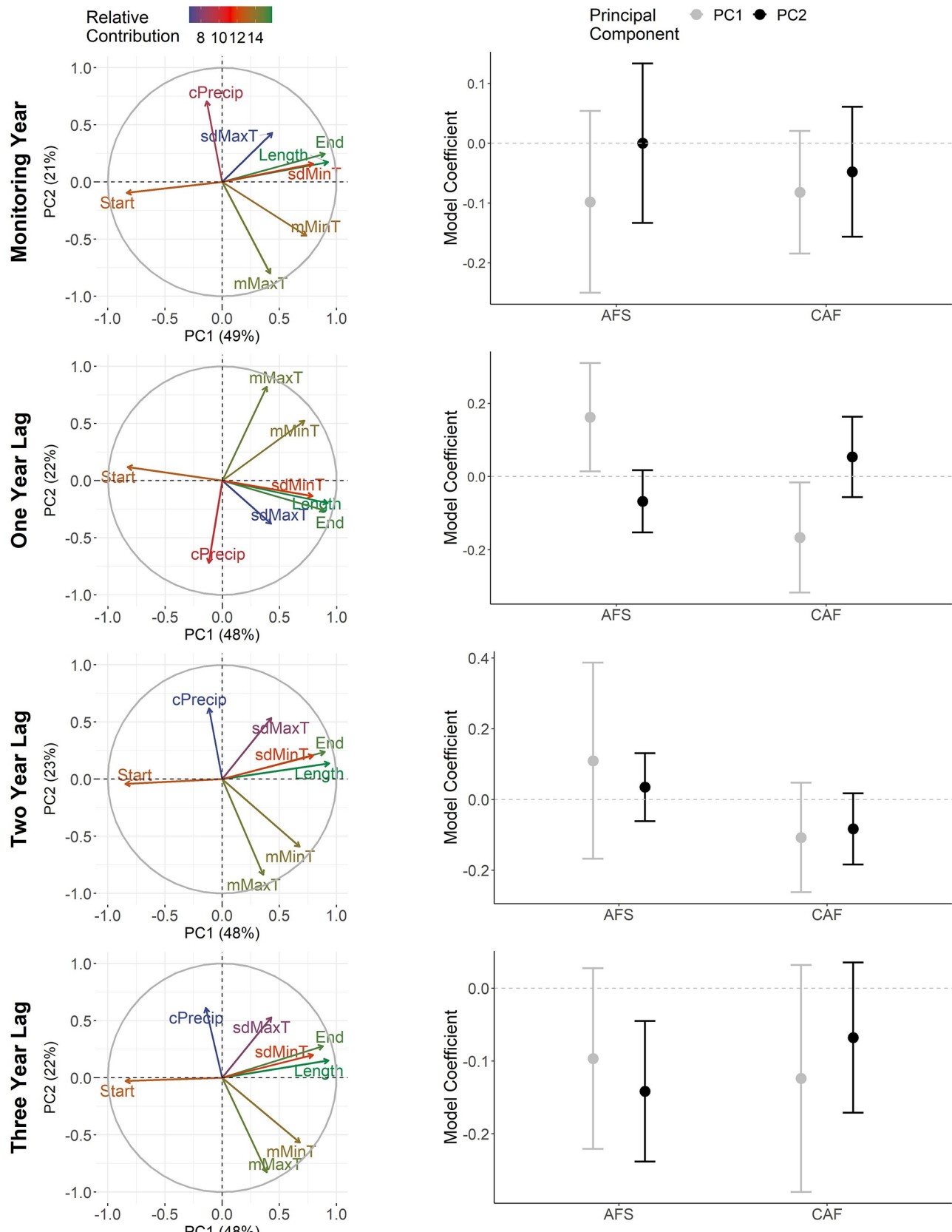

**Fig 2. Relationships between the demography of woodland caribou and principal components describing the meteorological growing season.**
Left-column graphs depict the relative contribution of meteorological variables (Start = season start, End = season end, Length = season length, mMaxT = mean maximum temperature, sdMaxT = standard deviation maximum temperature, mMinT = mean minimum temperature, sdMinT = standard deviation minimum temperature, cPrecip = cumulative precipitation) to the first two principal components. Right-column graphs show the coefficients (with 95% CIs) for each principal component for models evaluating effects on adult female survival (AFS) and calf: adult female (CAF) ratios. Rows reflect the time frame considered.

Across all demographic models using meteorological data, explained variation for the fixed effects was generally low (AFS models: all $R^2_{GLMM(m)} = \leq 0.12$; CAF models: all $R^2_{GLMM(m)}$ <0.03). For AFS models, much of the variation was explained by the addition of year- and population-specific random effects ($R^2_{GLMM(c)} = 0.98$–$0.99$). For CAF models, the additional variation explained by these random effects was lower ($R^2_{GLMM(c)} \leq 0.18$). Note that the variation explained by the year- and population-specific random effects will be constant across all seasonal demographic models—because the random-effects structure is held constant—and therefore we focus primarily on the variation explained by the fixed effects for the other seasonal analyses.

**3.2.2. Phenological growing season.** The first two PCs of the phenological growing season had the highest relative contribution from growing season length, the date of senescence, and iNDVI (Fig 3; S5 Appendix). Across all time periods, the first PCs described longer, more productive growing seasons. For the second PCs, seasonal descriptions varied by time period. For the monitoring year and two- and three-year lags, the second PCs described growing seasons that were less productive, had later senescence and a delayed peak of NDVI values. For one year lags, the second PC described growing seasons that were productive but productivity peaked earlier and senescence occurred earlier.

Annual variation in phenological conditions had a greater influence on CAF than AFS (Fig 3). Across all time periods, phenological conditions described by the first two PCs had no correlation with AFS as all 95% CIs strongly overlapped zero. CAF, in contrast, showed relatively strong correlations with variation in phenological conditions on a one year lag. With the first PC for this time period, which described longer, more productive growing seasons, CAF had a positive correlation ($\beta = 0.09$, 95% CI: 0.01, 0.17). This relationship yielded a change in predicted CAF from 0.08 to 0.17 as growing seasons lengthened and became more productive within the range of phenological conditions modelled (Fig 5). With the one-year lagged second PC, which described growing seasons that peaked early in productivity and had early senescence, CAF had a negative correlation ($\beta = -0.15$, 95% CI: -0.25, -0.06). This relationship yielded predicted CAF values that declined from 0.27 to 0.07 across the modelled phenological conditions (Fig 5). Population-specific coefficients for both one-year lagged PCs did show latitudinal trends, although the direction of correlation differed as the first PC showed a negative correlation (scaled $\beta = -0.01$, 95% CI: -0.02, 0.00) and the second PC had a positive correlation (scaled $\beta = 0.03$, 95% CI: 0.00, 0.06; Fig 6). In general, phenological models performed similarly to those developed with meteorological variables as explained variation by the fixed effects was low across all models for both demographic rates (AFS models: $R^2_{GLMM(m)} \leq 0.09$, $R^2_{GLMM(c)} = 0.98$–$0.99$; CAF models: $R^2_{GLMM(m)} < 0.02$, $R^2_{GLMM(c)} \leq 0.17$).

**3.2.3. Snow season.** The first two PCs of the snow season had a relatively even contribution from the 11 input variables (Fig 4; S5 Appendix). For the first PCs, seasonal descriptions varied by time period as the monitoring year and first two lags generally described shorter snow seasons with low snowfall whereas three-year lags described longer snow seasons with more snowfall. Second PCs primarily characterized snow seasons in terms of temperature. Across all time periods, the second PCs generally described colder seasons with increased temperature variability.

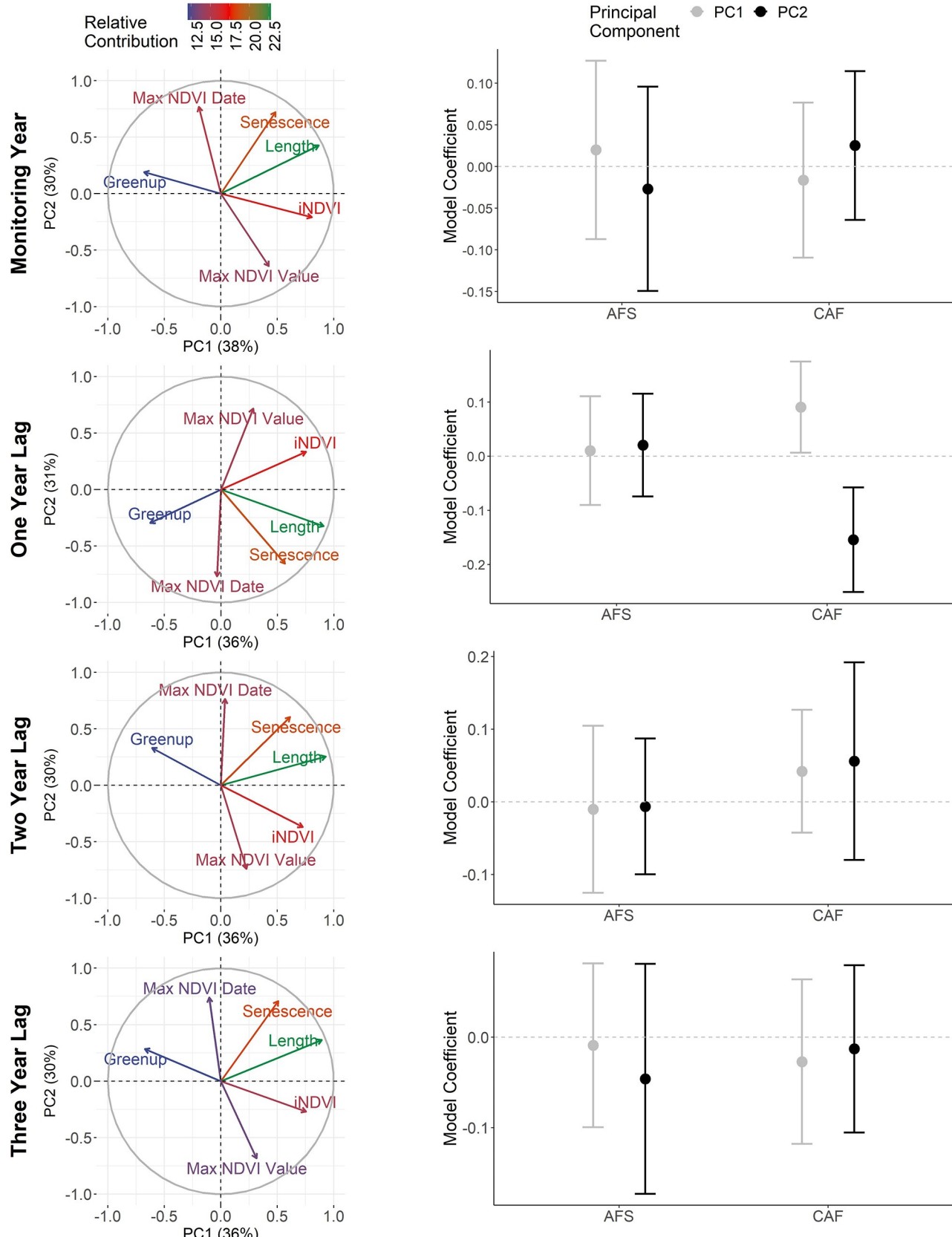

**Fig 3. Relationships between the demography of woodland caribou and principal components describing the phenological growing season.** Left-column graphs depict the relative contribution of phenological variables (note: length = season length) to the first two principal components. Right-column graphs show the coefficients (with 95% CIs) for each principal component for models evaluating effects on adult female survival (AFS) and calf: adult female (CAF) ratios. Rows reflect the time frame considered.

Temperature variation during the snow season appeared to have a greater influence on the two demographic rates than variation in season length and snow accumulation (Fig 4). The first PC, which described season length and relative snow accumulations, had no strong correlations with AFS during any time period. CAF trended toward a weak negative correlation with shorter snow seasons and low snow accumulation with this correlation most pronounced during the birth year ($\beta$ = -0.06, 95% CI: -0.13, 0.01). CAF also had a positive correlation with longer snow seasons and higher snow accumulations on a three-year lag ($\beta$ = 0.08, 95% CI: 0.00, 0.16). Population-specific coefficients for these CAF relationships did not show a latitudinal trend ($p$>0.56). In contrast to the first PC, both demographic rates had correlations with the second PCs, which described colder seasons with increased temperature variability, though the correlation direction and the time period varied between rates. AFS had a strong negative correlation with these colder seasons during the monitor year ($\beta$ = -0.16, 95% CI: -0.27, -0.05) and when they occurred on three-year lag ($\beta$ = -0.15, 95% CI: -0.24, -0.05). For both relationships, predicted values of AFS declined by 15% (monitoring year: 0.95 to 0.80; three-year lag: 0.94 to 0.79) as seasons became colder and more variable within the range of modelled values (Fig 5). CAF showed an opposite relationship, trending toward a positive correlation with these type of snow seasons with the most pronounced correlation occurring on a two-year lag ($\beta$ = 0.13, 95% CI: 0.03, 0.23). Predicted values of CAF from the two-year lag model increased from 0.06 to 0.20 as seasons became colder and more variable (Fig 5). Among these stronger correlations with the second PC, a latitudinal trend in population-specific coefficients was only evident for the correlation with AFS during the monitoring year ($\beta$ = -0.13, 95% CI: -0.24, -0.02; Fig 6). Similar to the other seasonal analyses, explained variation in AFS and CAF by the fixed effects was low for all demographic models developed for the snow season (AFS models: $R^2_{GLMM(m)} \leq 0.17$, $R^2_{GLMM(c)} = 0.98$–$0.99$; CAF models: all $R^2_{GLMM(m)} < 0.02$, $R^2_{GLMM(c)} \leq 0.16$).

## 4. Discussion

Ungulate population dynamics are known to be impacted by climatic variation, often mediated by population density [1, 2, 5]. Low density populations should be buffered from climate effects due to greater per-capita nutritional resources, but this pattern did not hold for woodland caribou, which generally occur at low densities. Instead, we found some relatively strong demographic responses to annual climatic variation, including both adult females and juvenile life-history stages. Although CAF and AFS responded to climate variability in different ways, our results suggest that the buffering effect of low population density may be context-specific, with factors other than population density playing an equal or greater role in mediating climate effects.

The relatively strong influences of climatic variation on the demography of woodland caribou that we found may be attributable, at least in part, to two interacting factors: predation and the life-history traits of woodland caribou. Much of the previous research investigating the interaction of climatic variation and population density on ungulate demography has been conducted in environments devoid of natural predators or where predation impacts are minimal (e.g., [2, 5, 61]). In these environments, individuals have minimal trade-offs when accessing resources. If climatic conditions are poor but the population density is low, demographic

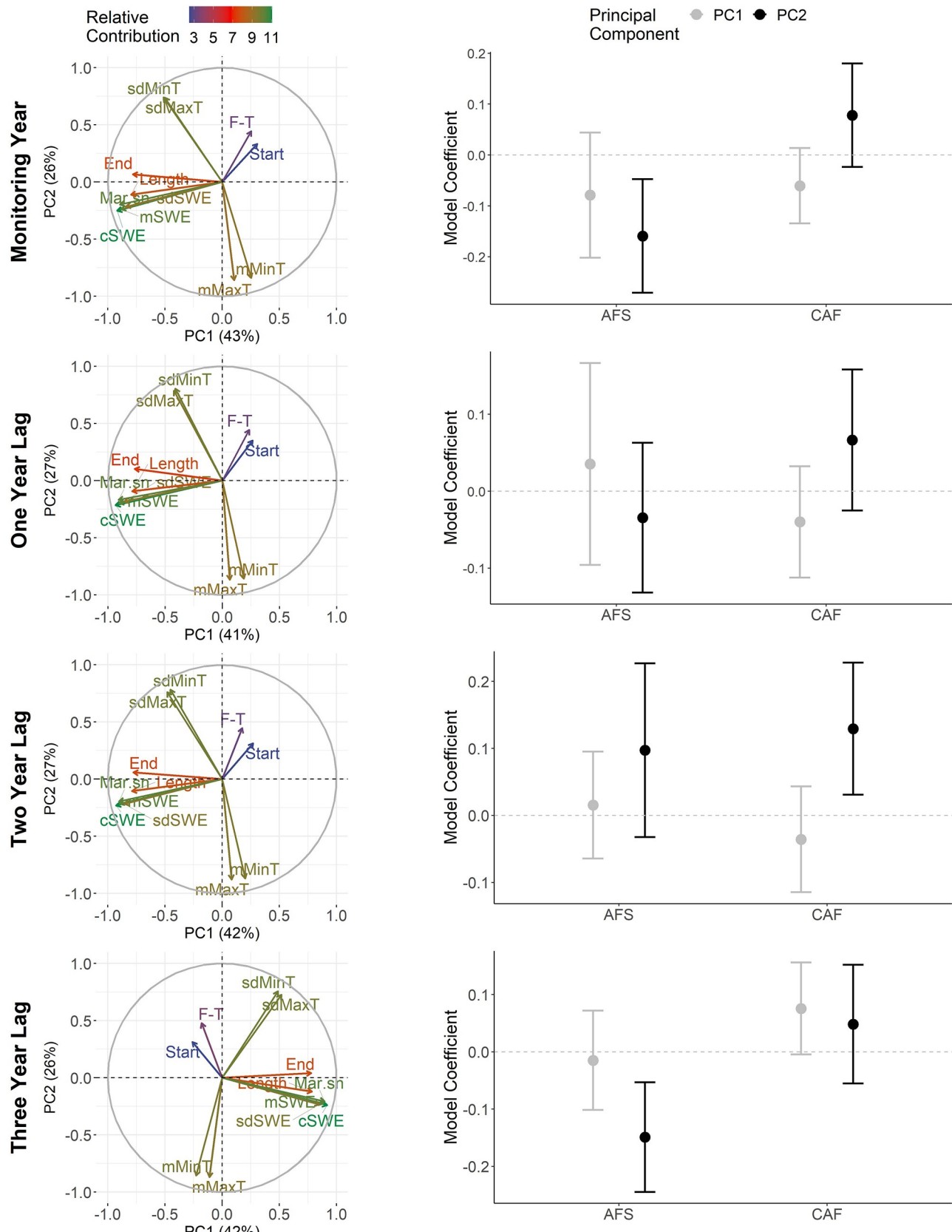

**Fig 4. Relationships between the demography of woodland caribou and principal components describing the snow season.** Left-column graphs depict the relative contribution of snow season variables (Start = snow cover start, End = snow cover end, Length = season length, Mar.sn = March cumulative snow, F-T = freeze-thaw events, mMaxT = mean maximum temperature, sdMaxT = standard deviation maximum temperature, mMinT = mean minimum temperature, sdMinT = standard deviation minimum temperature, cSWE = cumulative snow water equivalents, mSWE = mean snow water equivalents, sdSWE = standard deviation snow water equivalent) to the first two principal components. Right-column graphs show the coefficients (with 95% CIs) for each principal component for models evaluating effects on adult female survival (AFS) and calf: adult female (CAF) ratios. Rows reflect the time frame considered.

impacts may be minimal because individuals can still find sufficient resources by searching larger areas. In our study, all populations of woodland caribou reside in areas with low primary productivity and intact predator guilds, and predation is thought to be limiting [43, 44]. These caribou reduce predation risk by spatially separating from other sympatric ungulates (e.g., moose [*Alces alces*]), which typically occur in more productive landscapes, and their generalist predators [62]. This spatial segregation strategy, however, incurs trade-offs. Caribou have low reproductive rates compared to other sympatric cervids, and are also relatively easy to kill if they do encounter a predator [63]. Consequently, caribou are restricted to these low-productivity refugia, which is marginal habitat for other ungulates, and may not be able to expand their search area to obtain sufficient resources when climatic conditions reduce resources. For caribou "living on the edge" [64] in these refugia, even small changes in climatic conditions may result in resource reductions that are sufficient to elicit strong demographic impacts, even at low caribou densities.

Another factor potentially mediating the relatively strong climate effects on woodland caribou was the age structure of our study populations. Climatic variation can disproportionately

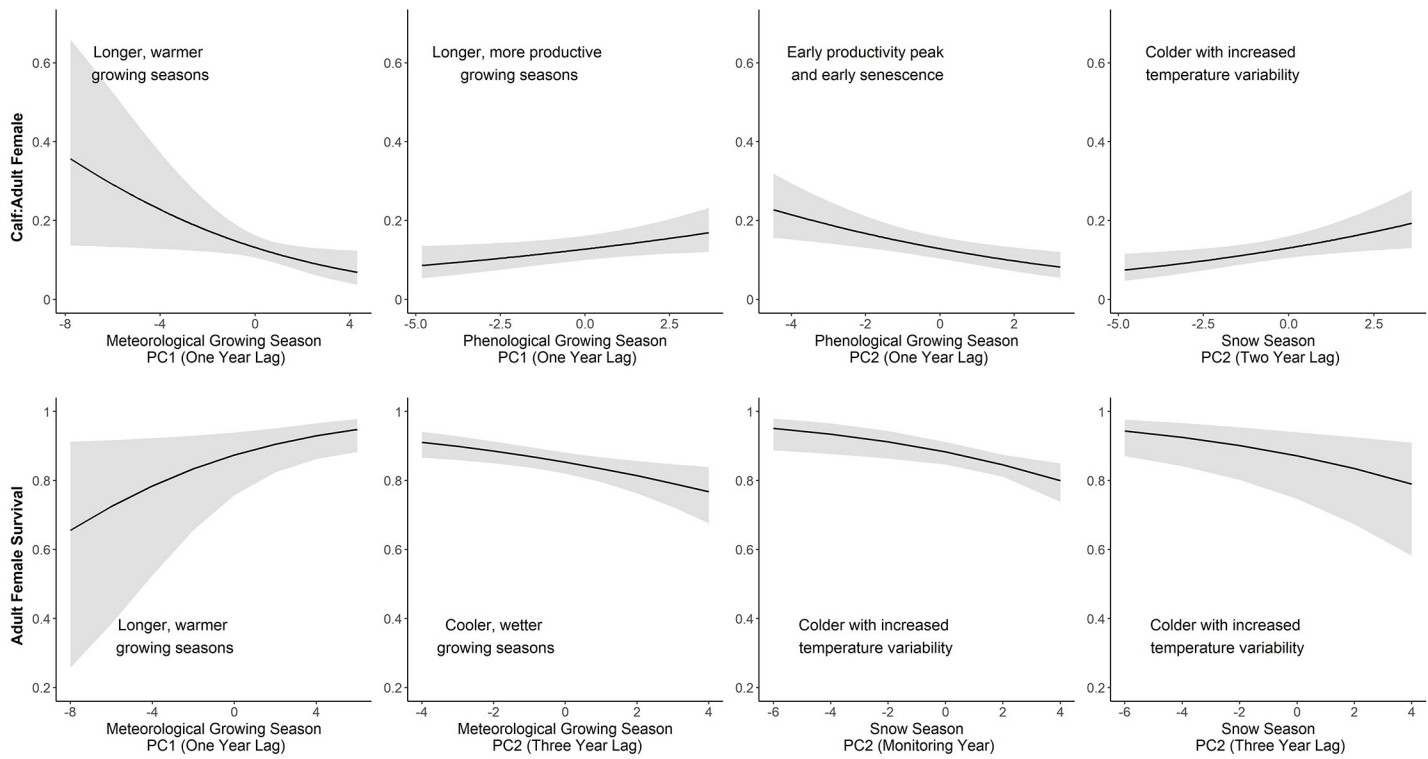

**Fig 5. Predicted responses (with 95% CIs) of caribou demographic rates (calf: Adult female ratios and adult female survival) to principal components (PCs) representing various meteorological and phenological variables.** Principal components shown are those with parameter estimates having 95% confidence intervals that did not overlap zero. For each facet, descriptions are provided for the referenced principal component. Note that x-axes vary among facets.

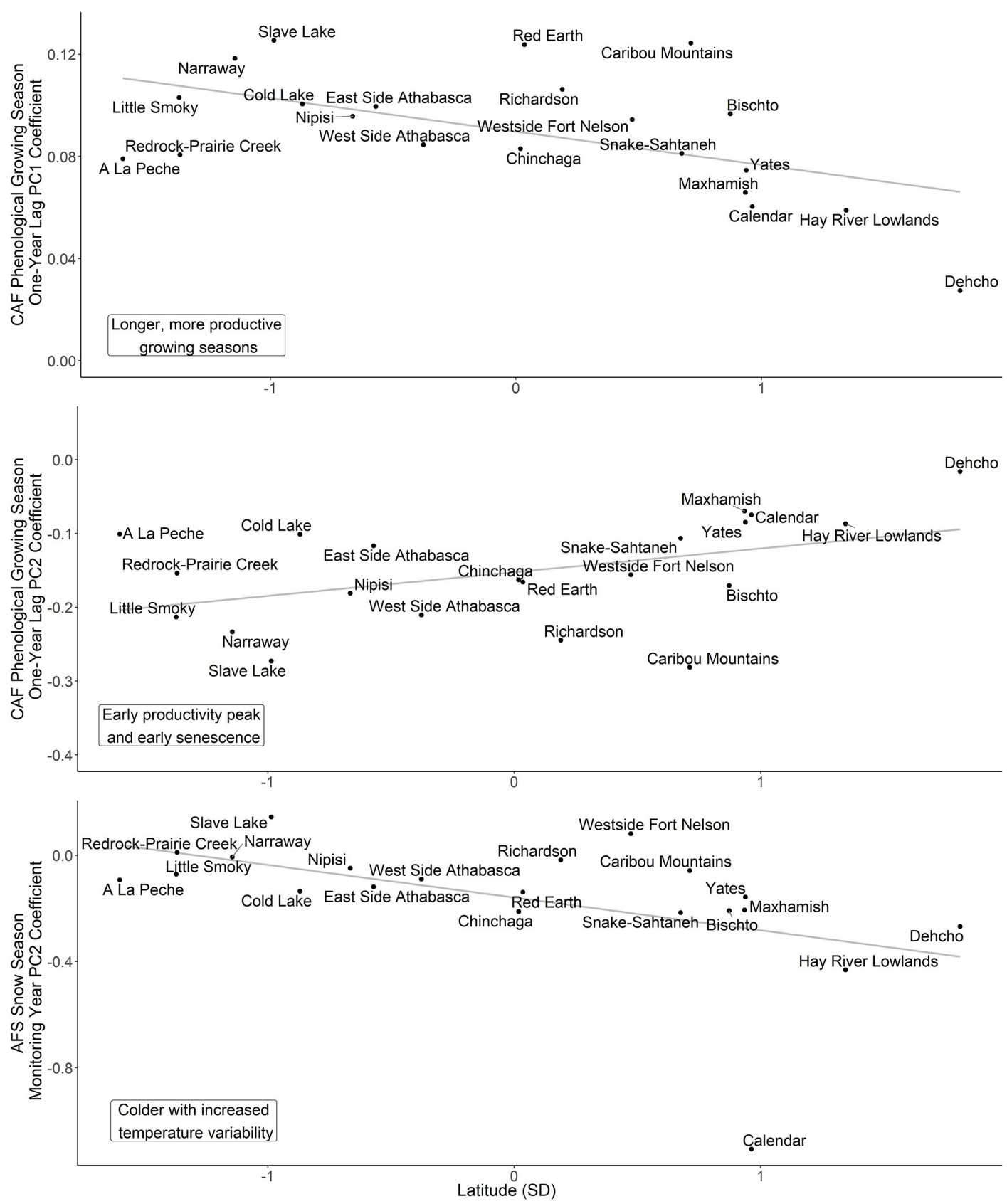

**Fig 6. Latitudinal trends in population-specific coefficients for the effects of principal component (PC) variables, which characterized meteorological and phenological seasons, on adult female survival (AFS) and calf: Adult female (CAF) ratios within 21 populations of woodland caribou in western Canada.** Latitudinal trends shown are those where: i) the PCs have a strong correlation with either AFS or CAF (95% CIs of model coefficients did not overlap zero); and ii) 95% CIs of coefficients for latitudinal trend did not overlap zero. Latitude has been standardized so one unit equals one standard deviation (SD; mean latitude = 57.41871˚, SD = 2.41999˚). Note that the y-axis differs among graph facets.

affect young and old age classes, producing disparate outcomes for populations at similar densities but different age structures [2, 7, 65]. Although the age structure of our study populations was unknown, most have chronically low rates of juvenile recruitment ([52]; S1 Appendix), suggesting an age structure skewed toward older age classes. Our finding that AFS negatively responded to adverse winter conditions during the monitoring year is consistent with an older age distribution [2], and continued low rates of juvenile recruitment could further accelerate declines in these already threatened populations.

## 4.1. Adult female survival

Climatic effects on AFS of woodland caribou were somewhat surprising, given the expectation that low-density populations should show weaker responses to climatic variation [2, 5] and that among vital rates, AFS should be the least sensitive [66]. Previous research on woodland caribou in eastern Canada found weak climatic effects on AFS (weak positive correlation with increasing snowfall; [29]). Our differing results underscore how demographic effects from climatic variation are likely to be context-specific. For example, differences in population age structure [2, 5, 7] and potentially greater climatic variation across our much larger study area could explain these differences. We also considered multiple time lags because energetic models for northern ungulates indicate that thresholds of energy accumulation or depletion that will ultimately impact survival can take years to cross [20].

AFS was most strongly correlated with meteorological conditions during winter, with lagged and unlagged effects. For northern ungulates, over-winter survival depends on energy reserves and depletion rate during winter [20]. Winter conditions can also influence predator-prey dynamics [7, 25]. During the monitoring year, AFS was negatively correlated with colder winters with high temperature variability, suggesting that these conditions may deplete energy reserves below critical levels and/or increase predation. Freeze-thaw events are a potential mechanism, and occurred more often during these winters due to increased temperature variability (S5 Appendix). Freeze-thaw events can negatively impact over-winter survival by decreasing resource availability [5, 34] and may increase predation by enhancing predator movement efficiency relative to their prey [67]. These mechanisms could also interact; for example, caribou in worse body condition following a freeze-thaw would be even more vulnerable to predation [7, 25].

Lagged climate effects on AFS were also evident, including both over-winter and growing season effects, with energy reserve accumulation and depletion as a likely mechanism. Longer, warmer growing seasons in the previous year were correlated with higher AFS, possibly due to a longer period of resource accumulation and/or a shorter period of energy depletion (e.g., a shorter ensuing winter). Wetter growing seasons, in contrast, were negatively correlated with AFS on a three-year lag, suggesting a long, slow decline in body condition eventually lowering survival in subsequent years. Possible mechanisms include reduced forage quality [68], increased prevalence of disease [69] or increased insect harassment [70]. These lagged climate effects also support the hypothesis that woodland caribou are energetically "living on the edge" within their low-productivity refugia, with any energy deficits incurred difficult to recover from in subsequent years. However, given that annual climatic conditions will vary spatially

and temporally, patterns of lagged effects on ungulate populations are likely to be context-specific (e.g., two-years lags could have stronger effects than three-year lags; [26]).

## 4.2. Juvenile recruitment

For ungulates, juvenile recruitment is more variable than adult survival, and is also more sensitive to climatic effects and resource limitation [17]. Consistent with this broad pattern, we found that juvenile recruitment was influenced by growing season conditions, particularly by plant phenology, and are likely mediated by maternal condition in the years prior to the monitoring (i.e., birth) year. This finding is consistent with previous research on *Rangifer* sp. as maternal condition affects pregnancy rates, fetal survival rates, and offspring body mass, all of which will impact subsequent juvenile recruitment [6, 20, 71]. In particular, late-growing-season conditions (i.e., autumn) had strong effects, consistent with findings for other ungulates in seasonal environments [19, 20, 72]. Juvenile recruitment in our study was positively correlated with longer, more productive growing seasons with delayed plant senescence in the autumn prior to birth, but declined during long, warm autumns in the previous year. Moreover, when growing seasons were shorter with an earlier peak in productivity, juvenile recruitment declined in the subsequent year. These complex responses could have several explanations. First, CAF ratios are sensitive to changes in both reproduction and adult female survival rates [73]. AFS in turn had a strong positive correlation with longer, warmer growing seasons on a one-year lag (Fig 5), suggesting increasing survival of older, less productive females that should be most sensitive to adverse climate conditions [2, 74]. Increased survival of senescent females could reduce CAF ratios if juvenile survival remains relatively constant. A second explanation is that the two modelled growing seasons were actually capturing different climatic characteristics. The one-year lagged meteorological and phenological growing seasons principal components had low correlation (all $r \leq |0.26|$; S6 Appendix), and therefore longer but warmer growing seasons may not equate to higher forage quality or abundance. Indeed, warmer, drier autumn meteorological conditions appeared to be correlated ($r = 0.26$ vs. $r = 0.24$) with an earlier peak in plant productivity and early senescence, likely indicative of declining forage quality and quantity in late summer and autumn. An example of this potential mismatch between growing season length and forage quality is provided by Paoli et al. [72], who showed that longer growing seasons may not be beneficial to the reproductive success of reindeer if the later end of such seasons are accompanied by a reduction in the quantity or quality of preferred forage.

Recruitment was also influenced by winter weather. In general, CAF ratios were positively correlated with lagged effects of longer snow seasons with increased snowfall, or colder snow seasons with increased temperature variability. Because longer and/or colder snow seasons are unlikely to positively affect maternal condition [20], trends in CAF ratios were likely driven by changes in AFS. Indeed, directional trends in CAF ratios were generally opposite of those observed for AFS. As suggested previously, lower AFS can result in higher CAF ratios, particularly if juvenile survival is less affected than survival of older females [74]. This pattern potentially provides further support for an older age distribution within many of our study populations. Moreover, in populations of woodland caribou that are stable, juvenile recruitment is positively correlated with less snowfall (i.e., opposite to our trend; [30]) or unaffected by snowfall [40].

## 4.3. Conservation implications in a changing climate

As a species adapted to arctic and subarctic climates, woodland caribou are expected to be increasingly impacted by climate change [75]. Now and in the future, the northward expansion and increasing abundance of apparent competitors (e.g., moose and deer) will be an important climate-related factor influencing caribou population dynamics [36, 76]. Here, we show that

caribou population dynamics may also be influenced by direct climate effects. For many of these effects, predicted effect sizes were relatively large (although explained variation was low —see *Limitations and Future Directions* below), which has implications for strategies aimed at stabilizing and recovering caribou populations. For instance, AFS declined by 16% as winters in the monitoring year became colder and more variable, at least across the full range of modelled conditions (Fig 5). Given CAF ratios associated with population stability (~29 calves per 100 females, assuming 85% AFS; [77]), then effects from these winter conditions on AFS could, on their own, influence whether populations are growing, stable or declining (population growth rate range: 0.92–1.09). These direct climate effects should be accounted for in addition to known stressors contributing to declines in many caribou populations (e.g., disturbance-mediated apparent competition; [78, 79]).

Our results indicate that direct climate effects must be considered and controlled for when testing management actions to conserve caribou. For example, wolf reduction has been deployed across a number of caribou ranges with varying demographic effects. In the Little Smokey range, CAF ratios increased 62%, on average, following six years of wolf reduction [80] and increased two-fold in the Aishihik range following five years of wolf reduction [81], but had limited or no effect in other ranges [82, 83]. In our study, CAF ratios were predicted to increase by 113% in response to differing growing season length and productivity in the year prior to birth (Fig 5), suggesting that climate effects could confound interpretation of the efficacy of predator control or other management actions if control populations with similar climate conditions are not concurrently monitored (e.g., [82]). The inclusion of control populations, however, does not necessarily overcome this potential confound as even nearby caribou populations may experience different climate conditions (e.g., [2, 32, 84]). Nevertheless, annual variation in climate effects should be considered when evaluating the efficacy of management actions.

As global climate change progresses, latitudinal trends in caribou demography can reveal how caribou will respond to future climate change. Two latitudinal trends were evident in our study: phenology changes affected southern populations more strongly, whereas colder and more variable winters affected northern populations more. The enhanced sensitivity of southern populations to plant phenology likely reflects how warming temperatures interact with precipitation to create substantial variation in forage conditions during late-summer and autumn along the southern edge of the boreal forest biome. As a capital breeder, caribou are sensitive to changes in forage conditions during this time [20]. Warm, wet years result in high, sustained forage productivity [85], whereas dry, warm years results in vegetative "browning" along the boreal forest's southern edge and poor forage conditions at the end of the growing season [85], creating differences in CAF ratios. Impacts from poor forage conditions may be exacerbated in southern populations by increased risk-sensitive foraging due to higher predator populations [78]. This relationship between changing forage conditions and caribou demography may portend deteriorating future conditions for northern populations, particularly if these conditions are accompanied by increasing populations of apparent competitors and their generalist predators.

For northern populations, the increased sensitivity to colder and more variable winters is likely due to an increase in freeze-thaw events (S5 Appendix), which are known to negatively impact ungulate demography [5, 34]. The intensity of icing following freeze-thaw events depends on temperature, rainfall amount and snowpack depth [86], and the trend of shallower snowpacks along the southern edge of the boreal forest may decrease the severity of freeze-thaw events, mitigating impacts on ungulate populations [86, 87]. However, the frequency of freeze-thaw events is predicted to increase with climate change, creating uncertainty regarding future impacts on populations of woodland caribou.

### 4.4. Limitations and future directions

Although we documented some relatively strong climate effects on the demography of woodland caribou, explained variation across all models was low, indicating that other factors have a higher influence on CAF and AFS than climatic variation. A growing body of research indicates that the degree of landscape disturbance (natural and human-caused) within caribou range is the most important driver of caribou declines [35, 37, 77, 79]. A recent analysis by Johnson et al. [79] showed that the proportion of disturbance within caribou range explained 54% of the variation in CAF ratios across 58 caribou populations. Future analyses that include climate-disturbance interactions are a critical next step, because climate can work synergistically with disturbance to facilitate the expansion of other apparent competitors (e.g., white-tailed deer) into caribou range, a dynamic that appears to result in increased predation of caribou [36, 88].

Our modelling framework focused on season-specific as well as lagged effects, but did not include cumulative effects due to sample size limitations. However, cumulative effects of climate variability, including those occurring within a year (i.e., a "bad" growing season followed by a "bad" snow season) or across years (e.g., two or more successive "bad" summers) are likely to have strong effects on caribou. As data accumulates on this threatened species [37], future analyses could include these cumulative climate effects.

Finally, we further caution that our analyses focused on linear relationships between climatic variation and caribou demography, but thresholds undoubtedly exist and will be revealed as climate change continues. Such climate thresholds are to be expected given that a species' distribution is often dictated by a climate envelope [89] and these thresholds may not be static, often interacting with density-dependent factors [33]. Continued monitoring of caribou-climate relationships, particularly across significant climate gradients (i.e., at the trailing edges of climate envelopes; [90]), should therefore be an integral part of conservation strategies aimed at stabilizing and recovering these threatened populations.

## Supporting information

**S1 Appendix. Caribou demographic data.**
(DOCX)

**S2 Appendix. Delineating seasons.**
(DOCX)

**S3 Appendix. Diagnostic testing of demographic models.**
(DOCX)

**S4 Appendix. R packages.**
(DOCX)

**S5 Appendix. Expanded results of principal component analyses.**
(DOCX)

**S6 Appendix. Correlations among meteorological and phenological variables.**
(DOCX)

## Acknowledgments

We thank Marco Apollonio, Luca Corlatti, and one anonymous reviewer for providing helpful feedback on an earlier version of this manuscript. We also extend thanks to the Governments of Alberta, Northwest Territories, and British Columbia for providing the caribou

demographic data. We are grateful to Youngwook Kim and John Kimball from the University of Montana for providing data on freeze-thaw events in our study area.

## Author Contributions

**Conceptualization:** Craig A. DeMars, Sophie Gilbert, Robert Serrouya, Stan Boutin.

**Data curation:** Allicia P. Kelly, Nicholas C. Larter, Dave Hervieux.

**Formal analysis:** Craig A. DeMars.

**Funding acquisition:** Robert Serrouya.

**Methodology:** Craig A. DeMars, Sophie Gilbert, Robert Serrouya.

**Supervision:** Robert Serrouya, Stan Boutin.

**Writing – original draft:** Craig A. DeMars.

**Writing – review & editing:** Craig A. DeMars, Sophie Gilbert, Robert Serrouya, Allicia P. Kelly, Nicholas C. Larter, Dave Hervieux, Stan Boutin.

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
