## [Decision Letter · Decision Letter 0]

27 May 2021

PONE-D-21-12789

Demographic responses of a threatened, low-density ungulate to annual variation in meteorological and phenological conditions

PLOS ONE

Dear Dr. De Mars

Thank you for submitting your manuscript to PLOS ONE. After careful consideration, we feel that it has merit but does not fully meet PLOS ONE’s publication criteria as it currently stands. Therefore, we invite you to submit a revised version of the manuscript that addresses the points raised during the review process.

As pointed out by one reviewer the lenght of the discussion is actually excessive so I encourage you to shorten it trying to focus on your main results and their outcomes moreover in the methods I suggest you to better describe how you evaluated  the survival estimates which currently lack a proper description.  

We look forward to receiving your revised manuscript.

Kind regards,

Marco Apollonio

Academic Editor

PLOS ONE

Journal Requirements:

3) We note that you have stated that you will provide repository information for your data at acceptance. Should your manuscript be accepted for publication, we will hold it until you provide the relevant accession numbers or DOIs necessary to access your data. If you wish to make changes to your Data Availability statement, please describe these changes in your cover letter and we will update your Data Availability statement to reflect the information you provide.

Reviewers' comments:

Reviewer's Responses to Questions

**Comments to the Author**

1. Is the manuscript technically sound, and do the data support the conclusions?

Reviewer #1: Yes

Reviewer #2: Yes

2. Has the statistical analysis been performed appropriately and rigorously? 

Reviewer #1: Yes

Reviewer #2: Yes

3. Have the authors made all data underlying the findings in their manuscript fully available?

Reviewer #1: Yes

Reviewer #2: Yes

4. Is the manuscript presented in an intelligible fashion and written in standard English?

Reviewer #1: Yes

Reviewer #2: Yes

5. Review Comments to the Author

Reviewer #1: I’ve critically reviewed the MS of DeMars et al. on the demographic responses of woodland caribou to climatic/ecological variables, which the authors submitted to PLoS ONE. The role of a reviewer is to spot the weak points of a MS but, frankly, I did not find any major weakness in this MS (with few exceptions, see comments below). The introduction is well written, the authors provide the reader with sufficient details to understand the issue at stake; hypotheses and predictions are well outlined. Perhaps – but this is largely a matter of personal taste – one thing I found a bit disappointing is an excessive focus on north-American ungulates. There is plenty of research on ungulates (cervids, but not only) in the northern hemisphere, that perhaps deserves to be mentioned to offer a broader perspective. The modeling part is also properly done, and I have just provided some small suggestions (see details below).

The only major limitation of the study is its length, especially in the Discussion: I acknowledge and appreciate the intent of the authors to provide a compelling analysis of the results, but 10 pages of discussion is way too much. Long texts distract the attention of the reader from the main point you’re trying to make, and dilutes the juicy part of your work. If there is anything I can suggest to improve this otherwise excellent MS, is to trim the text (especially in the M&M and in the Discussion).

Specific comments:

l. 4. & 66: Add latin name after caribou

l. 146-148: OK, but can you provide more details about this sample of marked individuals (E.g., age, numbers…) and the methodological approach? I miss details as to the methods used, to estimate mortality, and could not find relevant information in the supplementary (not sure if it’s my fault or not. Incidentally, “Supplemental Material A” does not exist, it’s “S1_Appendix”, please fix this inconsistency).

l. 163-245: I found this part overly lengthy, if there is a way to trim the text a bit (perhaps moving some explanations to the supplementary), I think the reading flow would benefit

l. 250-252: I understand the logic of this structure, but I wonder if you actually need this model complexity: have you tried compare this random structure with simpler structures?

l. 268-270: isn’t this a repetition of l. 250-252? Consider reshuffling to avoid redundancies

l. 285: please mention where this scaling formula comes from. If I’m not wrong, the reference should be Smithson & Verkuilen (2006)

l. 287: please add a reference for Beta regression modeling (e.g. Ferrari & Cribari-Neto 2005)

l. 310-311: I think I missed how you tested the GOF of the model; did you do residual diagnostics with quantile residuals?

Hope this helps!

Kind regards,

Luca Corlatti

Reviewer #2: I found this ms sound and well founded by data and detailed statistical analyses. In my opinion the ms strongly underlines that theoretical models need validation based on large datasets covering many populations and/or large part of the range of a species. Regarding the results/conclusions the context specificity is a crucial one. The large range and the number of populations typically results in this kind of contradiction; here it is addressed and underlined well. I support the publication of the ms.

6. PLOS authors have the option to publish the peer review history of their article (what does this mean?). If published, this will include your full peer review and any attached files.

Reviewer #1: **Yes: **Luca Corlatti

Reviewer #2: No

---

## [Author Response · Author response to Decision Letter 0]

15 Jul 2021

Responses to Reviewers’ Comments

NOTE: All line numbers referenced in the following responses refer to line numbers in the revised manuscript.

ACADEMIC EDITIOR:

Comment AE.1: As pointed out by one reviewer the length of the discussion is actually excessive so I encourage you to shorten it trying to focus on your main results and their outcomes moreover in the methods I suggest you to better describe how you evaluated the survival estimates which currently lack a proper description."

Thank you for these suggestions. In our revised manuscript, we have edited the Discussion to reduce its length and sharpen the focus on our key results. The revised Discussion is now 2,431 words compared to 3,183 words in the original manuscript. We hope that this reduction improves the readability of the manuscript and we are confident that our key findings and inferences are maintained in this revised version.

Authors' Response: To better describe how survival estimates of adult females were calculated, we added the following additional information (lines 146–153):

“For AFS, monitoring data from VHF- or GPS-collared adult females (≥2 years old; exact ages on capture are unknown) in each population were used to derive estimates of annual survival rates (x ® = 33.7 adult females monitored/caribou range/year [range: 8–115]; see S1 Appendix for yearly estimates). For VHF-collared females, survival status was determined by aerial telemetry flights conducted 4–12 times per year (52,53), a monitoring frequency found to produce unbiased survival estimates (54). Annual rates of AFS for each population were estimated using the Kaplan-Meier method in a staggered entry design (52,53,55).”

These added lines also include three additional references that also describe how survival estimates were calculated.

REVIEWER 1:

Comment R1.1: Perhaps – but this is largely a matter of personal taste – one thing I found a bit disappointing is an excessive focus on north-American ungulates.

Authors' Response: We agree that citations are, to a degree, influenced by “personal taste”. That said, we endeavored to provide 1–2 citations that were most appropriate for a given statement. Given the growing literature on climate-ungulate relationships, more citations could be listed in places, but we focused on trying to be concise in our citations. That said, we do note that of the ~26 primary studies on ungulate-climate relationships we cite in our paper, 14 are from studies situated in Europe.

Comment R1.2: The only major limitation of the study is its length, especially in the Discussion: I acknowledge and appreciate the intent of the authors to provide a compelling analysis of the results, but 10 pages of discussion is way too much. Long texts distract the attention of the reader from the main point you’re trying to make, and dilutes the juicy part of your work. If there is anything I can suggest to improve this otherwise excellent MS, is to trim the text (especially in the M&M and in the Discussion).

Authors' Response: Thank you for this suggestion and, after revising our Discussion, we agree that in its original form the Discussion was too lengthy. As noted in our response to the Academic Editor, we have shortened the Discussion by ~750 words. We believe that this reduction increases the readability of the paper without diminishing our main findings and inferences. 

Comment R1.3: l. 4. & 66: Add latin name after caribou

Authors' Response: Thank you. We have the added the Latin name in the referenced locations (now lines 4 and 67).

Comment R1.4: l. 146-148: OK, but can you provide more details about this sample of marked individuals (E.g., age, numbers…) and the methodological approach? I miss details as to the methods used, to estimate mortality, and could not find relevant information in the supplementary (not sure if it’s my fault or not. Incidentally, “Supplemental Material A” does not exist, it’s “S1_Appendix”, please fix this inconsistency).

Authors' Response: Thank you for catching the misnaming of the appendices/supplementary material in the manuscript. This has been corrected throughout. 

As noted in our response to the Academic Editor, we have added additional information on survival estimation for adult females. We have also included 3 additional references for further information on the methods used. 

Comment R1.5: l. 163-245: I found this part overly lengthy, if there is a way to trim the text a bit (perhaps moving some explanations to the supplementary), I think the reading flow would benefit

Authors' Response: We agree with this suggestion. In the revised manuscript, the referenced section has reduced to 886 words from an original length of 1,166 words. Some of the removed information has been added to the S2 Appendix.

Comment R1.6: l. 250-252: I understand the logic of this structure, but I wonder if you actually need this model complexity: have you tried compare this random structure with simpler structures?

Authors' Response: This model structure was dictated by our data and study design. Throughout our analyses, we used model structures that explicitly maintained caribou population (or range for the geographical area) as the primary sampling unit. This structure generates variance estimates that reflect among-population variation. Because of the large geographical extent of our study area, we felt it was appropriate to generate population-specific estimates of trend rather than an overall global mean, which would obscure population-specific differences. We included the continuous autoregressive correlation structure because of potential autocorrelation in successive years of a particular climate-related variable.

Comment R1.7: l. 268-270: isn’t this a repetition of l. 250-252? Consider reshuffling to avoid redundancies

Authors' Response: Although these two lines seem related because they describe mixed-model structures, they are referencing different parts of the analysis and different models. Lines 250-252 (now lines 237-239) describe univariate mixed models that are assessing for temporal trends in the climate-related variables. Lines 268-270 describe the structure of generalized mixed models used to assess meteorological and phenological effects on caribou demography. For mixed models, it is common practice to describe the random- and fixed-effects structure of a given model. Because the random- and fixed-effects structure of the models differ between these two parts of the analysis, we feel that removing one line or the other would delete critical information as to how each part of the analysis was conducted. 

Comment R1.8: l. 285: please mention where this scaling formula comes from. If I’m not wrong, the reference should be Smithson & Verkuilen (2006)

Authors' Response: Good suggestion. We have added this citation (line 272).

Comment R1.9: l. 287: please add a reference for Beta regression modeling (e.g. Ferrari & Cribari-Neto 2005)

Authors' Response: We added this citation (line 274). Thank you.

Comment R1.10: l. 310-311: I think I missed how you tested the GOF of the model; did you do residual diagnostics with quantile residuals?

Authors' Response: Thank you for this comment. We have added additional information on the diagnostic procedures we used for all demographic models in the S3 Appendix (see the section “Assessing Goodness-of-Fit of Demographic Models” therein). Note that we also provide R2 for generalized linear mixed models, which is a summary statistic for quantifying goodness-of-fit, following Nakagawa and Schielzeth (2013) and Nakagawa et al. (2017). 

REVIEWER 2: No comments to address

---

## [Editor Report · Decision Letter 1]

20 Sep 2021

Demographic responses of a threatened, low-density ungulate to annual variation in meteorological and phenological conditions

PONE-D-21-12789R1

Dear Dr. DeMars

We’re pleased to inform you that your manuscript has been judged scientifically suitable for publication and will be formally accepted for publication once it meets all outstanding technical requirements. I am sorry for the delay in this answer due to personal problems, I make you my compliments for your well written and clear final version.

Kind regards,

Marco Apollonio

Academic Editor

PLOS ONE
---

## [Editor Report · Acceptance letter]

28 Sep 2021

PONE-D-21-12789R1 

Demographic responses of a threatened, low-density ungulate to annual variation in meteorological and phenological conditions 

Dear Dr. DeMars:

I'm pleased to inform you that your manuscript has been deemed suitable for publication in PLOS ONE. Congratulations! Your manuscript is now with our production department. 

Kind regards, 

on behalf of

Prof. Marco Apollonio 

Academic Editor

PLOS ONE